# Palm Tocotrienol-Adjuvanted Dendritic Cells Decrease Expression of the *SATB1* Gene in Murine Breast Cancer Cells and Tissues

**DOI:** 10.3390/vaccines7040198

**Published:** 2019-11-27

**Authors:** Sitti Rahma Abdul Hafid, Ammu Kutty Radhakrishnan

**Affiliations:** 1Malaysian Palm Oil Board, 6 Persiaran Institusi, Bandar Baru Bangi, Selangor 43000, Malaysia; 2Pathology Division, School of Medicine, International Medical University, Bukit Jalil, Kuala Lumpur 57000, Malaysia; Ammu.Radhakrishnan@monash.edu; 3Jeffrey Cheah School of Medicine and Health Sciences, Monash University Malaysia, Jalan Lagoon Selatan, Bandar Sunway 47500, Selangor Darul Ehsan, Malaysia

**Keywords:** palm tocotrienols, breast cancer, cancer therapy, special AT rich-binding protein 1 (*SATB1*), gene expression, immune response

## Abstract

The aim of this study was to evaluate the effectiveness of immunotherapy using dendritic cells (DC) pulsed with tumor lysate (a DC vaccine) in combination with daily supplementation of tocotrienol-rich fraction (TRF) to potentiate anti-tumor immune responses. We had previously reported that DC-vaccine immunotherapy together with TRF supplementation induced protective immunity to tumor challenge. Breast cancer was induced in female BALB/c mice. The mice were randomly assigned into the treatment groups. At autopsy, peripheral blood was collected in heparinized tube and the expression of cell surface molecules (CD40, CD80, CD83, and CD86) that are crucial for T-cell activation and survival were analyzed by flow cytometry. Tumor was excised from each animal and snap-frozen. Total RNA was extracted from each tumor tissue for microarray and gene expression analysis. Total protein was extracted from tumor tissue for protein expression studies using Western blotting. The results show that systemic administration of 1 mg TRF daily in combination with DC-vaccine immunotherapy (DC + TL + TRF) caused a marked reduction (*p* < 0.05) of tumor size and increased (*p* < 0.05) the survival rates of the tumor-inoculated mice. The expression of CD40, CD80, CD83, and CD86 were upregulated in peripheral blood from the DC + TL + TRF group compared to other groups. In addition, there was higher expression of FasL in tumor-excised mice from the DC + TL + TRF group compared to other groups. FasL plays an important role in maintaining immune privilege and is required for cytotoxic T-lymphocyte (CTL) activity. Microarray analysis identified several genes involved in the regulation of cancer. In this study, we focused on the special AT rich binding protein 1 (*SATB1*) gene, which was reported to have dual functions, one of which was to induce aggressive growth in breast cancer cells. Tumors from DC + TL + TRF mice showed lower (*p* < 0.05) expression of *SATB1* gene. Further study will be conducted to investigate the molecular functions of and the role of *SATB1* in 4T1 mammary cancer cells and DC. In conclusion, TRF supplementation can potentiate the effectiveness of DC-vaccine immunotherapy.

## 1. Introduction

Efforts to find a cure for cancer have been ongoing for some time. Researchers have used several approaches, including developing vaccines against some tumors such as breast, lung, colon, and prostate cancers [1,2,3,4,5,6,7]. Dendritic cells (DC) represent a key cell used to develop cancer vaccines as this cell is the most potent antigen-presenting cell (APC) [8,9,10], playing a key role in mediating anti-tumor immune responses. These cells possess unique properties that allow them to elicit primary and boost secondary immune responses as well as regulate the type of T cell-mediated immune responses [11,12,13,14,15]. In recent years, several studies have demonstrated that tumor antigen-pulsed DC (DC vaccines) can induce activation and proliferation of cytotoxic T-lymphocytes (CTL) and T-helper (Th) cells via antigen presentation by major histocompatibility complex (MHC) class I and class II molecules, respectively, to mediate anti-tumor responses [1,16]. However, in a number of studies involving human cancer patients and mouse models of cancer, DC-based vaccines only exhibited minimal effectiveness against established tumors [2,10,17,18,19]. Some studies have shown that addition of adjuvants such as cytokines [20] or micronutrients [21] can boost anti-cancer immune responses initiated by the administration of DC vaccines.

Tocotrienol-rich fraction (TRF) is the vitamin E fraction extracted from palm oil, which contains 70% mixed tocotrienols and 30% alpha-tocopherol [22,23]. Tocotrienols and tocopherols are fat-soluble vitamins related to the vitamin E family. Tocopherols can be found in several vegetable oils such as soybean oil, cotton seed oil, and sunflower seed oil, whilst tocotrienols are primarily found in palm oil, rice-bran oil, and cereal grains such as wheat, barley, and rice. Tocotrienols have been the focus of increasing research interest in the last 5–10 years as a unique nutritional compound. The scientific evidence to date has shown that in addition to being powerful biological antioxidants, tocotrienols can reduce cholesterol levels in people with hypercholesterolemia, may slow down the progression of atherosclerosis, and possess anti-cancer properties [22,23,24]. We have previously reported that vaccinating mice with DC pulsed with tumor lysate from 4T1 cells (DC + TL) and daily supplementation with 1 mg TRF inhibited tumor growth and produced a tumor-specific immune response [21,24].

The aim of the present study was to investigate the effectiveness of TRF as an adjuvant in enhancing immune response to prevent tumor growth using a syngeneic mouse model of breast cancer to elucidate the mechanism(s) of action using microarray and quantitative PCR approach.

## 2. Materials and Methods

### 2.1. Mice

Five to eight-week-old inbred female BALB/C mice were purchased from the Experimental Animal Facility of the Institute for Medical Research (IMR), Kuala Lumpur. The animals were housed at the same facility and were given a commercial pellet diet and water ad libitum. The cage bedding was changed every three days. The mice were allowed to acclimatize in their environment and housed under a 12 h light/12 h dark cycle in an air-conditioned room with temperature set at 25 °C. This study was conducted in accordance with international animal ethics guidelines and was approved by the Research and Ethics Committee of the International Medical University (IMU), Kuala Lumpur (Ref: 4.9/55/2007 dated 26 June 2007).

### 2.2. Generation of Bone Marrow-Derived Dendritic Cells

The bone marrow (BM) was harvested under aseptic conditions. The tibia and femur bones were obtained from euthanized BALB/c mice and the BM cells were obtained by flushing the cut tibia and femur bones with complete medium using a 22 cc needle syringe as described previously [19,24]. Complete medium (CM) consisted of RPMI 1640 media with 10% fetal bovine serum (FBS, heat-inactivated), 1% glutamine, and 1% penicillin–streptomycin (PenStrep) added. Then, the erythrocyte-depleted mouse BM cells were cultured in CM containing 10 ηg granulocyte–monocyte colony-stimulating factor (GM-CSF; Chemicon, Temecula, CA, USA) and 10 ηg/mL interleukin-4 (IL-4; Chemicon, Temecula, CA, USA) at 37 °C in a humidified incubator with 5% CO_2_ (Heraeus, Germany). At day 5–7, 20 ηg/mL of tumor necrosis factor-alpha (TNF-α, Chemicon, Temecula, CA, USA) were added to the culture to induce maturation in cells. The DC were usually harvested on day 7–9, depending on their maturation and differentiation, which were monitored using an inverted microscope. The DC were characterized by analyzing the expression of CD11c using flow cytometry (FAC Calibur, Becton-Dickson (BD) Biosciences, San Jose, CA, USA)

### 2.3. Cell Line

The tumor cell line used in this study was the 4T1 murine mammary cancer cells, purchased from the American Type Culture Collection (ATCC) (ATCC, Manassas, VA, USA). The 4T1 murine mammary cancer cells are spontaneously metastatic tumor cells from mammary gland tumor of BALB/C mice which are reported to be comparable to human stage IV breast cancer [25]. The 4T1 cells were initially cultured in a 25-mL culture flask (T25) in complete culture medium (RPMI 1640 media with 10% FBS, 1% glutamine, 1% PenStrep) as recommended by the ATCC. Once the cells were confluent, the cells were transferred into a 75-mL culture flask (T75).

### 2.4. Freeze–Thaw 4T1 Tumor Lysate

The 4T1 cells were cultured in the presence of 8 μg/mL TRF in T25 flasks overnight at 37 °C in a humidified incubator with 5% CO_2_ (Heraeus, Germany). The next day the 4T1 cells were collected into a 15-mL tube (Falcon, Atlanta, GA, USA) and harvested by centrifugation (1000 rpm for 5 min). The supernatant was discarded and the cells were resuspended in 1 mL culture medium. Tumor lysate (TL) from the 4T1 cells was prepared by subjecting these cells to several freeze–thaw cycles. The cells were first frozen in liquid nitrogen and then rapidly thawed at 65 °C. The freeze–thaw cycle was repeated 3–5 times. The cell lysates was centrifuged (2000 rpm for 5 min) and the supernatant was passed through a 30-μm nylon filter-column before it was aliquoted in vials and stored at −80 °C until use.

### 2.5. Preparation of Tumor Lysate-Pulsed DC

The spent culture medium from DC in a T25 flask was replaced with fresh medium and TL was added. The amount of TL added was based on a 3:1 ratio of confluent DC:TL. The flask was incubated at 37 °C in a humidified incubator with 5% CO_2_ for 24 h. Following this, culture supernatant was removed and the cells were washed thrice in Hanks’ Balanced Salt Solution (HBSS). The tumor-pulsed DC were recovered by centrifugation (2000 rpm for 3 min) and resuspended in 1 mL complete medium and left on ice prior to injection into mice.

### 2.6. Treatment of Mice

The six-week-old BALB/c mice were randomly assigned in to one of the five groups (control, vehicle, DC alone, DC + TL, and DC + TL + TRF) used in this study (Table 1). Each group consisted of three BALB/c mice. The mice in the three groups receiving DC injections (DC alone, DC + TL, and DC + TL + TRF) received weekly intra-peritoneal (i.p.) injections of DC for three consecutive weeks (Table 1). In addition, the mice in the DC + TL + TRF group were fed daily with 1 mg of TRF by oral gavage. A week after the last DC injection, the mice in all five study groups received a single injection of 4T1 cells (50 µL of 4T1 cells in phosphate-buffered saline (PBS)) into their mammary fat pad. As reported previously, the tumor was usually palpable on day 14 [21,24]. Tumor growth was monitored once a week by measuring the size of the primary tumor using a digital caliper. At the end of the study, all the mice were sacrificed. At autopsy, blood, tumor, and various organs were collected for further analysis.

### 2.7. Analysis of Cell Surface Markers on in Blood Mice

Whole blood (500–800 µL) was collected into heparin tubes. The red blood cells (RBC) were lysed using a commercial RBC lysis buffer according to the manufacturer-recommended protocol (Thermo Scientific, Altrincham, UK). Briefly, two drops of RBC lysis buffer were added to each tube containing the mice blood. The tubes were incubated at room temperature for 5 min. The lysis reaction was stopped by addition of PBS pH 8.0 solution. The tubes were centrifuged (1000× *g* for 5 min at 4 °C). The supernatant was discarded and the PBS wash step was repeated twice. Following this, the cells were stained with fluorochrome-conjugated antibodies against some mouse antigens, such as CD40-FITC (Miltenyi Biotec Inc., Auburn, CA, USA), CD80-PE (Miltenyi Biotec Inc., Auburn, CA, USA), CD83-FITC (Miltenyi Biotec Inc., Auburn, CA, USA), and CD86-PE (Miltenyi Biotec Inc., Auburn, CA, USA), for 30 min on ice. Then, the cells were washed with buffer (PBS with 0.1% bovine serum albumin (BSA)) and recovered by centrifugation (1800 rpm for 5 min). The cells were fixed by addition of 500 µL of wash buffer, followed by 500 µL of fixing solution (1% of paraformaldehyde solution). Each sample was prepared with unstained cells for comparison with stained cells. The tubes were thoroughly mixed before they were analyzed using flow cytometry using the Cell-Quest software provided by the manufacturer (FAC Calibur, Becton-Dickson (BD) Biosciences, San Jose, CA, USA). The population of interest was gated using the forward scatter (FSC) and side scatter (SSC) dot plot. For each acquisition, 10,000 events were acquired for data analysis. The acquired data was analyzed using the Cell-Quest software. For each sample, the percentages of cells stained with the fluorochrome-conjugated antibodies (FL1 (FITC/green fluorescence) versus FL2 (PE/red fluorescence)) in the gated population were obtained. The changes in the FSC and SSC channel were adjusted and compensated so that the populations were centralized on the dot plot.

### 2.8. RNA Extraction, Characterization and Integrity

For the microarray analysis, total RNA was extracted from tumors of mice (*n* = 3) from three groups (DC alone, DC + TL, and DC + TL + TRF). Total RNA was also extracted from 4T1 cells for this analysis. Total RNA was extracted using the TRI-reagent solution according to the manufacturer’s instructions (Molecular Research Center, Inc., Cincinnati, OH, USA). The concentrations of the extracted RNA and ratio of absorbance at 260 nm to 280 nm (A260/A280 ratio) were measured using the NanoDrop ND-1000 spectrophotometer (NanoDrop Technologies, Wilmington, DE, USA). The integrity of the extracted RNA samples was evaluated with the RNA integrity number (RIN) for each sample using the Total RNA 6000 Nano Kit with the Agilent 2100 Bioanalyzer (Agilent Technologies, Waldbronn, Germany). The RIN describes a gradual scale of RNA integrity from 1 (RNA completely degraded) to 10 (RNA without degradation). In general, a RIN that is higher than seven is accepted to be optimal in most of experiments. (http://www.biomedicalgenomics.org/The_RNA_Integrity_Number.html)

### 2.9. Sample Preparation and Hybridization for Microarray Study

Experiment was conducted using the ILLUMINA Beadchip array with MouseRef-8 v1.1 expression beadchips. The beadchip targets approximately 25,600 well-annotated RefSeq transcripts, over 19,100 unique genes, and enables the interrogation of eight samples in parallel. The MouseRef-8 Bead Chip content is derived from the National Center for Biotechnology Information Reference Sequence (NCBI RefSeq) database (Build 36, Release 22), supplemented with probes derived from the Mouse Exonic Evidence Based Oligonucleotide (MEEBO) set as well as exemplar protein-coding sequences described in the RIKEN FANTOM2 database. The microarray facilities were provided by Malaysian Genome Institute (MGI), University Kebangsaan Malaysia (UKM), Bangi, Malaysia. The complementary RNA (cRNA) synthesis and purification were performed using IlluminaTotalPrep RNA Amplification (Illumina, San Diego, CA, USA). All the RNAs were subject to cleanup using RNA cleanup (QIAGEN, Germantown, MD, USA) and hybridization steps followed with the manufacturer’s protocol.

### 2.10. Real Time RT-PCR

In this study, we used quantitative PCR (RT-PCR) to confirm the expression of special AT-rich binding protein 1 (*SATB1*) at the mRNA level. For this experiment, RNA from tumors of three animal groups (DC alone, DC + TL, and DC + TL + TRF) and 4T1 cells were used. The RNA samples treated with DNase treatment (Invitrogen Inc., USA) were used for the RT-PCR. The reaction mix consisted of 5 µL purified RNA (5 ng), 0.5 µL of enzyme mix, and 12.5 µL of 2 × reaction mix (SYBR Green Real Time PCR mix (Invitrogen Inc., Camarillo, CA, USA)); 10 µM of reverse and forward primers were added to the microwell plate. Ultra-pure water (Gibco, MD, USA) was added to each microwell to make the final volume 25 µL. The reaction was run at 50 °C for 2 min and 95 °C for 10 min, followed by 40 cycles at 95 °C for 15 s and 55 °C for 30 s, and an extension phase for 1 cycle at 95 °C for 60 s, 60 °C for 60 s, and 95 °C for 60 s (ramp time, 19.59 min) in a thermocycler (IQ5 thermocycler, Bio-Rad, Hercules, CA, USA). Triplicate samples were run for each RNA. The primers for murine beta-actin (*β-actin*) and *SATB1* genes were purchased from Invitrogen (Camarillo, CA, USA) and R&D Research (Minneapolis, MN, USA), respectively. The primer sequences are as follows: (a) *β-actin* (forward: 50-AGAAGGATTCCTATGTGGGGG-30, reverse: 50-CATGTCGTCCCAGTTGGTGAC-30) and (b) *SATB1* (forward: ACACAGCTC TGCTGCCCAAGCC, reverse: GACCAGCTGAGGACTG ATCGG). Data were normalized relative to the housekeeping gene.

### 2.11. Protein Expression

At autopsy, tumor was excised from all the experimental animals. Total protein was extracted from tumor tissue using a commercial protein extraction kit using the manufacturer’s recommended protocol (Milipore, USA) for Western blot analysis. Protein concentration of each sample was determined using the Bradford method (BioRad, Hercules, CA, USA). Protein extracted (30 ng) from each sample was resolved on 10% SDS-PAGE gels and electro-blotted onto nitrocellulose membrane using a semi-dry transfer system (Bio-Rad, Hercules, CA, USA). The membrane was incubated overnight at 4 °C with 5% blocking agent (GE Healthcare Life Sciences, Amersham PI, Chalfont, Amersham, UK) in wash buffer (0.1% (v/v) Tween 20 in PBS). After the blocking step, membrane was probed with primary antibody (SATB1 or FasL) (Novus Biologicals, USA) for 2 h at room temperature. Subsequently, the membranes were incubated for another two hours at room temperature with a secondary antibody, horseradish peroxidase (HRP) anti-rabbit (Novus Biologicals, Centennial, CO, USA). Detection was performed using an ECL detection kit according to the manufacturer’s instructions (GE Healthcare Life Sciences, Amersham PI, Chalfont, Amersham, UK). Following this, the membranes were stripped off from the probed antibodies, blocked, and re-incubated with mouse monoclonal GAPDH (R&D Research, USA) followed by HRP-labeled anti-mouse secondary antibody (R&D Research, Minneapolis, MN, USA). Bands on autoradiography films (GE Healthcare Life Sciences, Amersham PI, Chalfont, Amersham, UK) were quantified using ImageJ software and normalized against GAPDH. The list of primary and the corresponding secondary antibodies used for the Western blotting assay is listed in Table 2.

### 2.12. Statistical Analysis

Statistical analysis was done using the Student *t*-test and SPSS version 20 for the most of data. For the tumor incidence data, statistical analysis was done using the one-way ANOVA with post hoc Tukey test for multiple comparisons (SPSS version 20). The IMAGE J software (http:rsb.info.nih.gov/ij/index.html) was used for the densitometry analysis of protein bands whilst the Bead Studio 2.1 and GENE SPRING 7.0 were used for microarray analysis. Most of the data represent the ± SD of triplicate measurements. The quantitative PCR data represent mean value with ± SEM of the triplicate measurements.

## 3. Results

### 3.1. Anti-Tumor Effects of DC Pulsed with Tumor Lysate and Daily TRF Supplementation

The anti-tumor effect was assessed using a syngeneic mouse model of breast cancer. Previously, we had reported that the combination of DC pulsed with TL (DC + TL) and TRF supplementation showed marked anti-tumor effects, as a marked inhibition of tumor growth and metastasis was observed in the mice that received this treatment [21,24]. In the present study, we compared the effectiveness of un-pulsed DC (DC alone), DC pulsed with TL tumor lysate (DC + TL) and DC + TL with daily TRF supplementation (DC + TL + TRF) in preventing the growth of 4T1 cells in these mice. The DC treatment was given once a week for three consecutive weeks. Tumor was inoculated one-week after the last DC injection. In this model, tumor was usually palpable on day 14 after tumor injection [21,24].

A marked anti-cancer effect (*p* > 0.05) was observed in mice in the DC + TL + TRF groups (Figure 1).

DC pulsed with tumor lysate and oral supplementation of TRF in tumors in BALB/c mice revealed a significant anti-tumor effect (Figure 1). As shown in Figure 1, all mice in the control group developed tumor around week 3 and the tumor grew rapidly. A similar effect was observed in the mice from the vehicle group, which fed daily with vehicle, which was soy oil. Although there was a slight increase in tumor growth in mice from the three DC groups, their tumors were smaller when compared to tumors from the control or vehicle groups (Figure 1). Amongst the three DC-treated groups, the mice from the DC + TL + TRF group showed the slowest tumor growth. Furthermore, only one out the three mice in the DC + TL + TRF group developed tumor (Figure 1). These findings strongly suggest that DC + TL + TRF treatment method was most effective in preventing tumor growth compared to the other approaches.

### 3.2. Expression of CD40, CD80, CD83, and CD86 in Peripheral Blood Leucocytes

The expression of CD40, CD80, CD83, and CD86 on the surface of peripheral blood leucocytes (PBL) of mice was determined as these molecules are reported to be important surface markers that provide co-stimulatory signals required for T-cell activation and survival [24,25,26]. Generally, there was lower expression of CD40, CD80, CD83, and CD86 in the PBL obtained from mice from the control and vehicle groups compared to PBL from mice from the DC, DC + TL, and DC + TL + TRF groups (Figure 2). There was a marked increase (*p* < 0.05) in the number of PBL that expressed CD40, CD80, CD83, or CD86 in PBL from the DC + TL + TRF groups compared to the other groups (Figure 2). The expression of CD40 in the DC + TL + TRF group was 40% compared to the DC + TL (35%), DC (18%), vehicle (4%), and control (3%) groups (Figure 2). For CD80, the expression of the DC + TL + TRF group was 27% compared to DC + TL (24%); similar expression found in the DC and vehicle groups (16%), and the control group showed about 17% (Figure 2). For CD83, the expression was 39% in the DC + TL + TRF group, compared to 26% in the DC + TL group, 7% in the vehicle group, and 10% in the control group, respectively (Figure 2). For CD86 expression, the expression was also found to be increased in treatment group with TRF compared to the others (DC + TL + TRF: 37%, DC + TL: 20%, DC: 15%, vehicle group: 18%, and control group: 10%) (Figure 2).

### 3.3. Expression of Fas Ligand in Tumor Tissues

The Fas/Fas ligand (FasL) is reported to be one of the process required for the maintenance of immune privilege [27,28,29]. It has been reported that the CTL-mediated lysis of susceptible target is through the engagement of *Fas* by *FasL* expressed on the activated CTL [28,29] In this study, we looked at the changes in the regulation of Fas expression in tumors excised from the various study groups. The expression of FasL in tumors from the DC + TL + TRF group was almost 20% higher when compared to tumors from the other DC-treated groups (DC + TL: (13%); DC alone: (10%)) (Figure 3). In contrast, the expressions of FasL were relatively low in the vehicle (1 %) and control (1%) groups (Figure 3).

### 3.4. Microarray Analysis

We have conducted microarray experiment to view the expression levels of any genes within all treatments. Only RNA samples with RIN that was equal to or higher than 7 were used in this experiment (figure not shown). In order to reduce non-specific binding errors, the normalization intensity was conducted on all samples and groups. A dendogram (Figure 4a) was created with the data using the Bead studio 2.1. software. The analysis shows that the gene expression pattern obtained from the cultured 4T1 cells was quite distinct from the DC-treated groups, even though the 4T1 cells were used to induced tumor in these animals (Figure 4a). However, tumors excised from the three DC-treated groups (DC alone, DC + TL, or DC + TL + TRF) appear to show similar gene expression patterns. Furthermore, it was noted that there was a close resemblance in gene expression pattern between the DC + TL and DC + TRF as they appeared to cluster very closely in the dendogram (Figure 4a). The same pattern was observed in the Condition Tree analysis (Figure 4b), which was generated using an analysis software in Gene Spring 7.1. This analysis was performed to study the differences (if any) of gene expression between each treatment, which was then compared to the results from the DC + TL + TRF group. From the 25,600 reference genes annotated in this beadchip mouse array, only 18,118 genes could be used for the comparison studies (Figure 4b). The data from the microarray study was also analyzed using a volcano plot, which was generated by comparing the data from different treatments. The volcano plots (Figure 4c) were generated based of differential gene expression that was higher or lower than two-fold or more (≥2), with confidence interval set at 95% (*p* < 0.05) or 99% (*p* < 0.01). For this analysis, we focused on studying the difference in gene expression in tumors from the DC + TL and DC + TL + TRF groups as these treatment groups showed very similar genes expression pattern in the dendogram (Figure 4a) and Condition Tree (Figure 4b) analysis. In the volcano plot analysis, when the confidence level was set at 95% (*p* < 0.05) for, there were four genes that were differentially regulated (Figure 4c). The expression of *CRIP2* was up-regulated, whilst three genes (*IFITM1, MRC1,* and *SATB1*) were down-regulated (Table 3). When the confidence level was increased to 99% (*p* < 0.01), only one gene was differentially regulated (Figure 4c). This gene, *SATB1,* was found to be markedly (*p* < 0.01) down-regulated in tumors from the DC + TL + TRF group when compared to tumors from the DC + TL groups (Table 4).

### 3.5. Verification of Microarray Results by Quantitative PCR

To verify the findings from the microarray analysis, the expression of *SATB1* gene in the tumor tissues extracted from the three DC-treated groups (DC alone, DC + TL, and DC + TL + TRF) was analyzed using quantitative PCR (qPCR). Total RNA extracted from cultured 4T1 cells was also included in the analysis. A marked (*p* < 0.05) reduction in the expression of SATB1 gene was observed in tumors extracted from mice treated with DC + TL + TRF (Figure 5). This showed that there was consistency in results between the qPCR and microarray analyses. However, it should be noted that there were differences in the level of expression between qPCR and microarray analyses. In tumors from the DC + TL + TRF, the expression of *SATB1* was only 0.14-fold (Figure 5), whilst in the microarray analysis, the fold change noted was 0.396 (Table 4). Tumors from the DC + TL + TRF (0.14) as well as DC + TL (0.32) groups had significantly (*p* < 0.05) lower expression of the *SATB1* gene when compared to DC alone (0.59) or cultured 4T1 cells (1.0) (Figure 5).

### 3.6. Expression of SATB1 Protein

Western blotting analysis confirmed the results from the gene expression studies. There was a significant reduction in the expression of the *SATB1* protein expression in tumors from the DC + TL + TRF and DC + TL groups when compared to tumors from the control or vehicle groups (Figure 6a). When the densitometry analysis was performed, the results showed that the *SATB1* expression to be reduced in DC + TL + TRF (23.3% of relative density) group compared to other groups (Figure 6b). The control and vehicle groups showed almost similar expressions (100%, 98.8% of relative density) of the *SATB1* protein. The relative densities of *SATB1* protein expression in the DC alone and DC + TL groups were 60.22% and 47.99%, respectively (Figure 6b).

## 4. Discussion

DC-based cancer immunotherapy has become the main research interest in active immunotherapy [3,4]. For this approach to be successful, DC must be able to function well. This is a crucial step to produce an effective anti-tumor response, which contributes to the success of cancer immunotherapy. Prophylactic treatment with oral TRF supplementation together with DC in a mouse model of breast cancer was reported to inhibit tumor growth and induce tumor-specific immune responses [19,24]. The aim of the present study was to evaluate the effectiveness of immunotherapy using DC pulsed with tumor lysate (DC vaccine) in combination with daily supplementation of TRF from palm oil to potentiate anti-tumor immune responses. Several researchers have shown that TRF on its own can inhibit the proliferation of 4T1 murine mammary cancer cells using cell-based [30,31,32] and experimental [30,33] models. In addition, there are also numerous publications on the anti-cancer effects of TRF on various human breast cancer cells lines [30,31,32,33,34,35,36,37].

The results from this study demonstrated that daily oral TRF supplementation together with DC + TL injections can significantly inhibit the tumor growth and increase the rate of survival (Figure 1). Although previous studies have shown that oral TRF supplementation alone can inhibit tumor growth [33], the results when used in combination with DC immunotherapy were much better. The findings from the present study confirm previous report, which showed that the DC + TL + TRF treatment approach inhibited tumor growth and metastasis by promoting Th1 immune response through stimulating production of interferon-gamma (IFN-γ) and interleukin-12 (IL-12) [19,24,37].

The expression of CD40, CD80, CD83, and CD86 on the PBL of mice was performed as these molecules are reported to play key roles in antigen presentation to T-cells [25,26]. These molecules provide co-stimulatory signals that required for T-cell activation and survival. There was an increase in the expression of these cell surface molecules on PBL obtained from mice from the DC + TL + TRF group (Figure 2).

The expression of FasL was significantly increased in tumor tissues isolated from the DC + TL + TRF mice (Figure 3). This is an important finding, as expression of *FasL* is reported to be one of the important processes in the maintenance of immune privilege [24,25,26,27,28,29]. This protein is shown to have a crucial role in controlling cell death mechanism. Low expression of FasL can cause de-regulation in cancer cells, which is involved in immune escape mechanisms by the tumor cells. Lack of FasL expression is associated with poor prognosis of cancer patients [24,25,26,27,28,29].

Microarray analysis performed to compare gene expression in tumors from DC + TL and DC + TL + TRF flagged four differentially expressed genes. One gene (*CRIP2*) was upregulated, whilst three genes (*SATB1, MRC1,* and *IFITM1*) were down-regulated (Table 3) when the data were analyzed a using volcano plot with two-fold different changes and confidence interval at 95%. When a higher confidence level of 99% (*p* < 0.01) was used, only one gene (*SATB1*) was found in this category (Figure 4c). The *SATB1* gene, which codes for the special AT-rich binding protein 1, was markedly down-regulated (0.396-fold) in tumors from mice treated with DC + TL + TRF (Table 4). This *SATB1* gene was reported to have dual functions. For instance, Kohwi et al. [38] reported that SATB1, which is predominantly found in thymocytes, is responsible for the development of T-cells [38,39]. In this regard, the *SATB1* may be orchestrating the temporal and spatial expression of genes during T-cell development, especially in TH2 [38,39,40]. This is because researchers have shown that the *SATB1* protein functions to tether the specialized DNA sequences and bind to matrix attachment region (MAR), which are implicated in the loop domain organization of chromatin or chromatin remodeling [38,39,40]. In addition, aggressive breast cancer cells are found to express high levels of *SATB1* gene [39], which was found to be closely linked (*p* < 0.0001) with prognostic outcomes of cancer patients [40]. In cancer patients with poor prognosis, it was proposed that *SATB1* gene was able to reprogram chromatin organization and the transcription profiles of breast tumors, which promoted cancer growth and metastasis [40]. In the present study, the expression of SATB1 was markedly suppressed in tumors excised from mice treated with DC immunotherapy and supplemented with TRF (DC + TL + TRF).

The expression pattern of the *SATB1* gene correlated well with the protein expression in tumors excised from the experimental animals. As observed with the gene expression studies, there was marked suppression of SATB1 protein expression in tumors from mice treated with DC + TL + TRF (Figure 6).

## 5. Conclusions

In conclusion, this study has shown that TRF can be used to supplement DC-vaccine immunotherapy using an immunocompetent experimental model. The DC + TL + TRF approach appears to induce activation and promote survival of T-cells, as judged by the increased expression of CD40, CD80, CD83, and CD86 in PBL as well as FasL in the tumors. Increased expression of FasL protein in tumor tissue from the DC + TL + TRF-treated mice suggest that CTL activation may have taken place and this could have contributed to inhibition of tumor growth. One of the limitations of this study is the small number of mice used for the study. Also, the scope of the present study is rather narrow as we were not able to include more parameters to investigate. However, despite this the results are significant and we have managed to identify a key target protein that we could do more work on. *SATB1* was identified to be one of the potential targets that could have produced some of the anti-tumors effects observed. Further studies are being planned to elucidate the role of the *SATB1* gene in tumor growth and spread as well as to understand how TRF is able to modulate the expression of this gene.

## Figures and Tables

**Figure 1 vaccines-07-00198-f001:**
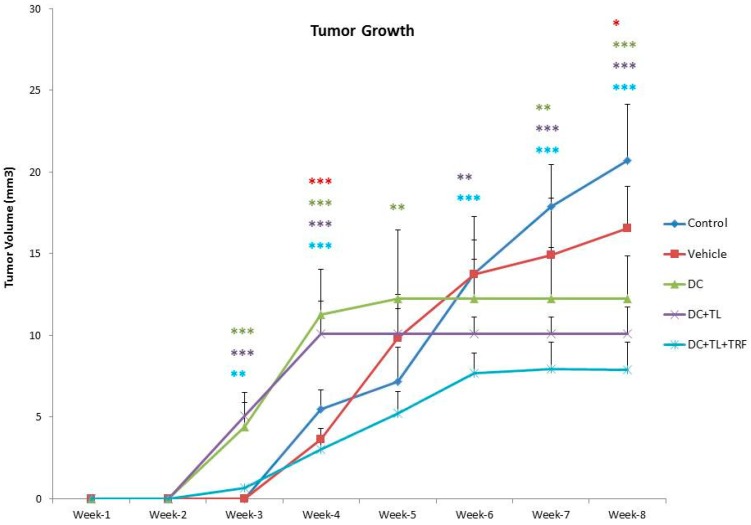
Tumor volume and tumor incidence in a mouse model. Tumor volume was measured every three days from the time tumor was palpable (day 14) until week 8, when all the animals were sacrificed (*n* = 3 per group). Animals in the control group did not receive any form of intervention; animals in the vehicle group were fed with soy oil; animals in the DC alone group received un-pulsed DC; animals in the DC + TL group received DC pulsed with tumor lysate from 4T1 cells; and the animals in the DC + TL + TRF group received DC + TL and daily supplementation of TRF. Data were analyzed using one-way ANOVA followed by the post-hoc Tukey multiple comparison test for pair-wise comparison. Results are shown as the mean ± SEM (* *p* < 0.05, ** *p* < 0.01, *** *p* < 0.001: significant compared to control group).

**Figure 2 vaccines-07-00198-f002:**
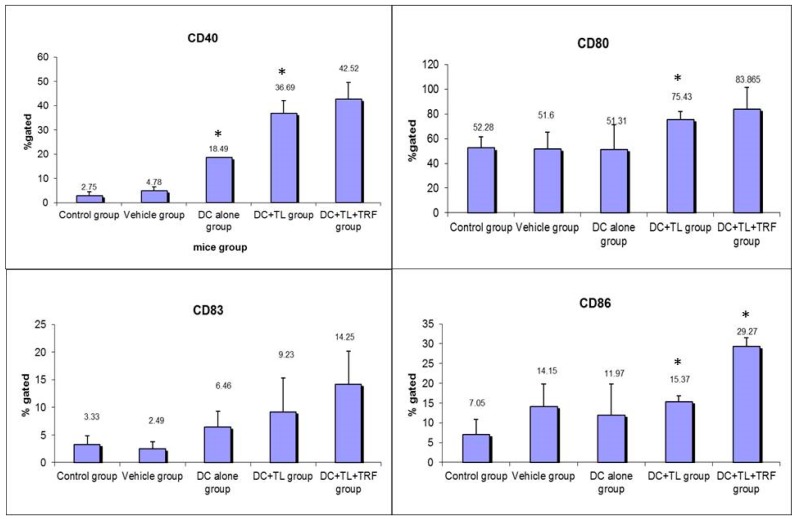
The expression of cell surface molecules (CD40, CD80, CD83, and CD86) on peripheral blood leucocytes obtained from control and experimental mice (*n* = 3 per group) at autopsy was stained with the relevant fluorochrome-conjugated antibodies and analyzed by flow-cytometry. For each sample, 10,000 cells were acquired. The results from the flow cytometer were analyzed using Student’s *t*-test for comparison with the expression in the control group. Control: tumor-induced but no treatment; Vehicle: tumor-induced and fed with soy oil; DC: tumor-induced and three injections of DC; DC + TL: tumor-induced and three injections of TL-pulsed DC; and DC + TL + TRF: tumor-induced, with three injections of TL-pulsed DC and daily TRF supplementation. * significantly different from the control group (*p* < 0.05).

**Figure 3 vaccines-07-00198-f003:**
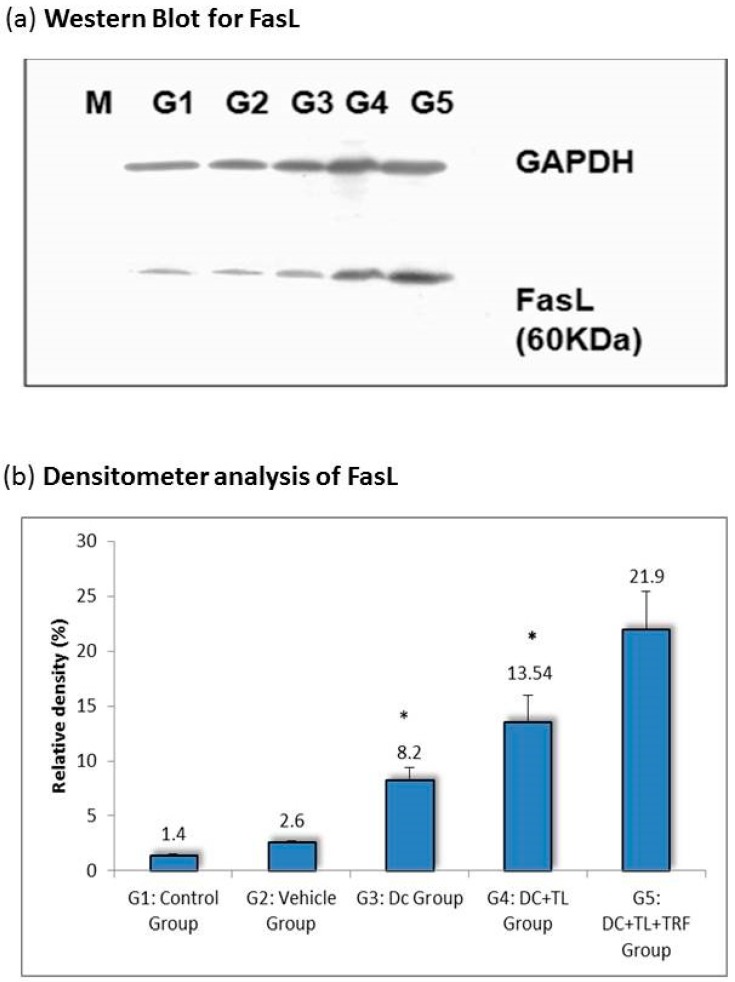
(**a**) A cell lysate preparation was prepared from the tumors harvested from the experimental mice as described before (*n* = 3 per group). Protein levels in the cell lysate were determined using the Bradford assay. Then, the cell lysate was subjected to SDS-PAGE, followed by Western blotting assay to estimate the level of FasL protein expression in these samples and then (**b**) the intensity of the expression was quantified using densitometry analysis (Image J, Software based on image processing program developed at National Institutes of Health). The results from the densitometer was analyzed using Student’s *t*-test for comparison with the expression in the control group. * denotes a significant difference from the control group (*p* < 0.05). M: Protein marker; G1: control group; G2: vehicle group; G3: DC group; G4: DC + TL group; G5: DC + TL + TRF group.

**Figure 4 vaccines-07-00198-f004:**
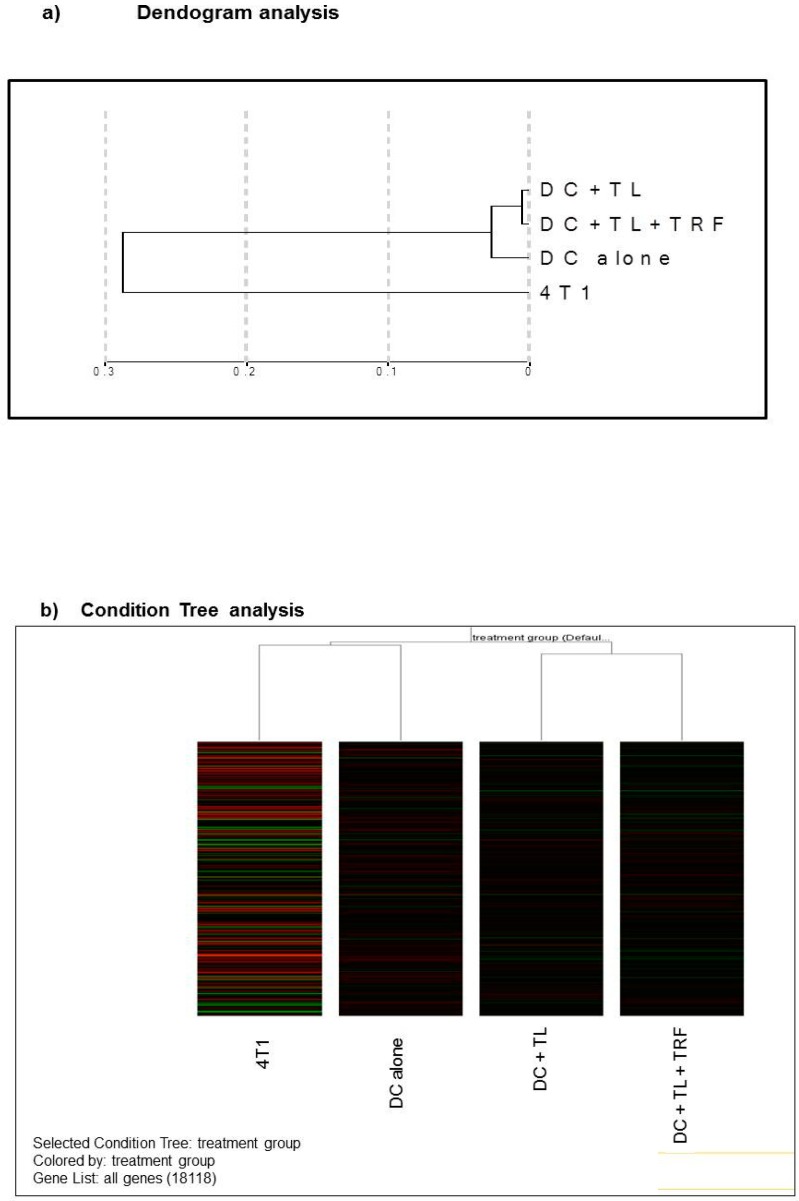
(**a**) Dendogram analysis showing clustering of similar samples (*n* = 3 per group). The dendogram was generated using Bead Studio 2.0 software. (4T1: RNA from 4T1 murine mammary cancer cells; DC alone: RNA from BM-derived DC; DC + TL: RNA from bone marrow (BM)-derived DC pulsed with tumor lysate (TL) from 4T1 cells; DC + TL + TRF: RNA from BM-derived DC pulsed with tumor lysate (TL) from 4T1 cells at 8 μg/mL). (**b**) Condition Tree analysis for all treatment groups (*n* = 3 per group). The most similar pattern was observed in DC + TL + TRF and DC + TL. There was a small difference between DC alone (untreated DC) and twice-treated DC (DC + TL and DC + TL + TRF). 4T1 showed a different pattern compared to the others. (**c**) Volcano plot analysis represent fold changes and confidence levels between two analyzed samples (*n* = 3 per group).

**Figure 5 vaccines-07-00198-f005:**
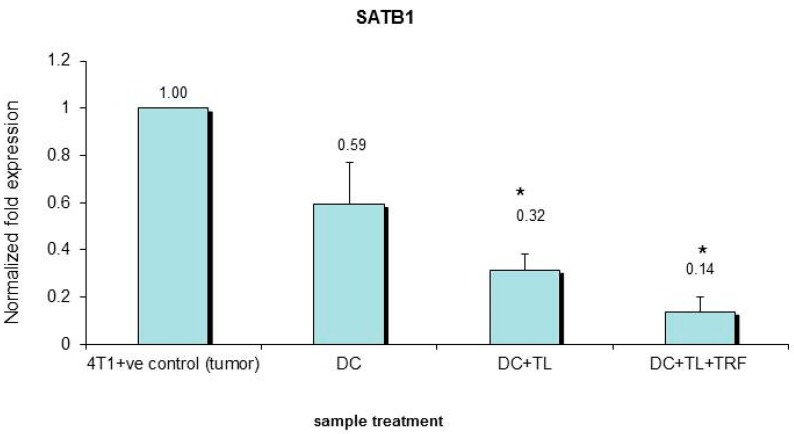
Validation expression of the *SATB1* gene in 4T1 cells and DC (treated and untreated) cells using quantitative real time PCR (qPCR). The qPCR assay was performed on RNA isolated from the 4T1 murine mammary cancer cells (**4T1**), untreated murine DC (**DC**), and DC treated with tumor lysate from 4T1 cells in the presence (**DC + TL + TRF**) or absence (**DC + TL**) of 8 μg/ mL TRF as described previously. The *β-actin* gene was used as a housekeeping gene and was used to normalize all qPCR data. Each biological replicate (*n* = 3 per group) was analyzed in triplicate. The qPCR results were analyzed using Student’s *t*-test to compare with the expression in the control group. (* denotes significantly different from 4T1 +ve control (tumor), (*p* < 0.05)).

**Figure 6 vaccines-07-00198-f006:**
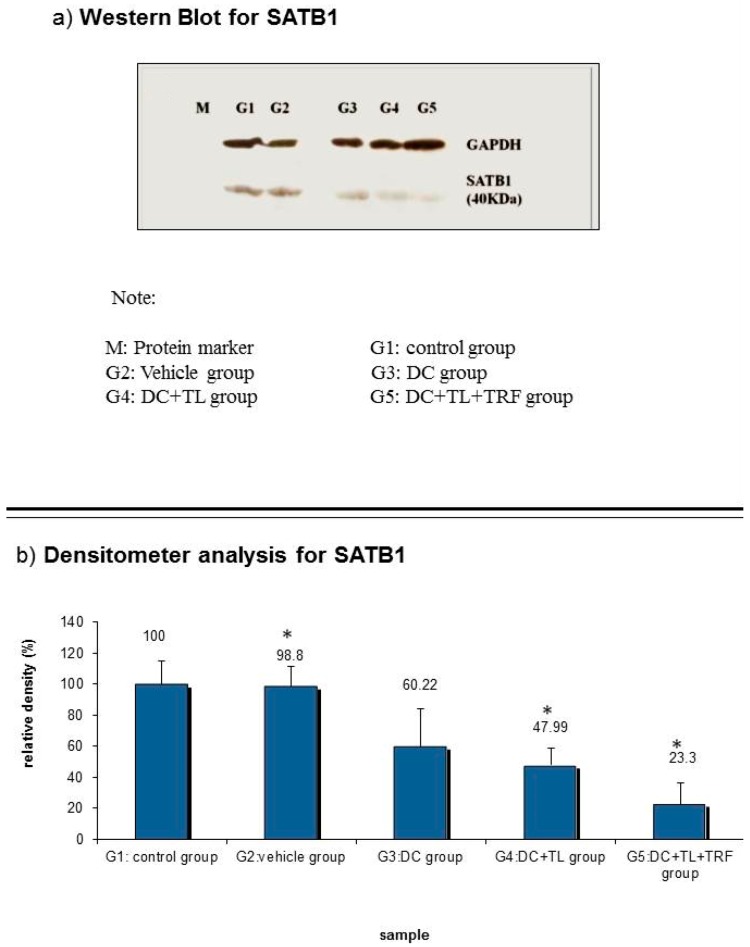
(**a**) A cell lysate preparation was prepared from the tumors (*n* = 3 per group) harvested from the experimental mice as described before. Protein levels in the cell lysate were determined using the Bradford assay. Then, the cell lysate was subjected to SDS-PAGE, followed by Western blotting assay to estimate the level of *SATB1* protein expression in these samples and then (**b**) the intensity of the expression was quantified using densitometry analysis (Image J). The results from the densitometer was analyzed using Student’s *t*-test to compare with the expression in the control group. (**G1:** tumor-induced without treatment; **G2:** tumor-induced and fed with vehicle; **G3:** tumor-induced and one DC injection once a week for three weeks; **G4:** tumor-induced and one DC + TL injection once a week for three weeks; **G5:** tumor-induced and one DC + TL injection once a week for three weeks and daily feeding with TRF). * Significantly different from control group (*p* < 0.05).

**Table 1 vaccines-07-00198-t001:** Study groups.

Groups	Name	Treatment
1	Control	Injected with 4T1 cells
2	Vehicle	Fed with vehicle daily Injected with 4T1 cells
3	DC	Injected with DC (three times) Injected with 4T1 cells a week after last DC injection
4	DC + TL	Injected with DC + TL (three times) Injected with 4T1 cells a week after last the DC + TL injection
5	DC + TL + TRF	Injected with DC + TL (three times) Fed with 1 mg TRF daily Injected with 4T1 cells a week after the last DC + TL injection

DC: dendritic cells; TL: tumor lysate; TRF: tocotrienol-rich fraction; 4T1: murine mammary cancer cells; Vehicle: 50 µL of soy oil.

**Table 2 vaccines-07-00198-t002:** List of primary and secondary antibodies used in the Western blotting assay.

No.	Primary Antibody	Secondary Antibody
(1)	Mouse FAS ligand/TNFSF6 antibody (AF526)	Goat IgG HRP-conjugated antibody (HAF017)
(2)	Human/mouse/rat GAPDH/G3PDH antibody, antigen affinity-purified polyclonal goat IgG (AF5718)	Goat IgG HRP-conjugated antibody (HAF017)

**GAPDH/G3PDH**: Glyceraldehyde 3-phosphate dehydrogenase; **HRP**: horseradish peroxidase; **IgG**: immunoglobulin G; **TNFSF6**: tumor necrosis factor super family member 6.

**Table 3 vaccines-07-00198-t003:** Gene profiling in two-fold different changes.

No.	Gene Name	Description	Regulation in DC + TL + TRF	Fold * Change
1.	*CRIP2*	*Mus musculus* cysteine-rich protein 2 (Crip2), mRNA	Up-regulated	2.337
2.	*IFITM1*	*Mus musculus* interferon-induced transmembrane protein 1 (Ifitm1)	Down-regulated	0.488
3.	*MRC1*	*Mus musculus* mannose receptor, C type 1 (Mrc1), mRNA	Down-regulated	0.471
4.	*SATB1*	*Mus musculus* special AT-rich binding protein 1	Down-regulated	0.396

* *p* < 0.05 in DC + TL + TRF vs. DC + TL.

**Table 4 vaccines-07-00198-t004:** Gene profiling in two-fold different changes.

No.	Gene Name	Description	Regulation in DC + TL + TRF	Fold * Change
1.	*SATB1*	Special AT-rich binding protein 1	Down-regulated	0.396

* *p* < 0.01 in DC + TL + TRF vs. DC + TL.

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
