# Peer review of "Palm Tocotrienol-Adjuvanted Dendritic Cells Decrease Expression of the SATB1 Gene in Murine Breast Cancer Cells and Tissues"

_vaccines, 2019, doi:10.3390/vaccines7040198_

Round 1
Reviewer 1 Report
The study of Hafid and Radhakrishnan aimed at evaluating the effectiveness of immunotherapy in combination with daily supplementation of tocopherol rich fraction on anti-tumor immune response in a murine model of breast cancer. The subject matter is of interest however several issues should be addressed.
Specific comments:
1) Materials and methods are incomplete: the number of animals for each experimental groups as well as the statistical analysis are lacking. To this respect the of use two-way ANOVA followed by Tukey-Kramer or Newman-Keuls as post-hoc test is recommended.
More details should also be added on flow cytometry analysis as:
the number of independent experiments that were performed during the course of the study and number of samples analyzed for each group. the methodology and parameters used for flow cytometry data acquisition (i.e. the equipment’s manufacturer, model, software, lasers and laser power settings, as well as the optical emission filters that were used for each fluorescent reagent) the sample size or number of cells counted in the population of interest and the minimum number of events for a target population.
In addition gating schemes must be addressed. It is also critical that authors include all information regarding controls that were used. These include unstained, biological, isotype, fluorescence minus-one or negative controls present within the cell suspension.
2) Results are often redundant and can be revised
3) Flow cytometry data must be also displayed as histogram or dot plot.
4) Results from qRT-PCR should be analyzed using the comparative DCt method
5) Study limitation should be discussed
6) Figures are of poor quality
Author Response
Response to Comments from Reviewer 1
The study of Hafid and Radhakrishnan aimed at evaluating the effectiveness of immunotherapy in combination with daily supplementation of tocopherol rich fraction on anti-tumor immune response in a murine model of breast cancer. The subject matter is of interest however several issues should be addressed.
Specific comments:
|
Comments |
Response from authors |
|
1) Materials and methods are incomplete: the number of animals for each experimental groups as well as the statistical analysis are lacking. |
· The materials and methods have been revised to address both these comments · Changes are shown in red font. |
|
To this respect the of use two-way ANOVA followed by Tukey-Kramer or Newman-Keuls as post-hoc test is recommended. |
· We have analyzed the data with one-way ANOVA with post-hoc Tukey · Changes are shown in red font. |
|
More details should also be added on flow cytometry analysis as: |
|
|
the number of independent experiments that were performed during the course of the study and number of samples analyzed for each group. |
· This information has been added to the methods section · Changes are shown in red font. |
|
The methodology and parameters used for flow cytometry data acquisition (i.e. the equipment’s manufacturer, model, software, lasers and laser power settings, as well as the optical emission filters that were used for each fluorescent reagent) |
· We have explained about how the gating was performed and the number of cells acquired for analysis. · The information on the equipment’s manufacturer, model, software has been mentioned in methods · Changes are shown in red font. · It is not possible to get the information about the lasers and laser power settings, as well as the optical emission filters that were used for each fluorescent reagent. |
|
The sample size or number of cells counted in the population of interest and the minimum number of events for a target population. |
· We have explained about how the gating was performed and the number of cells acquired for analysis. · Changes are shown in red font. |
|
In addition, gating schemes must be addressed. It is also critical that authors include all information regarding controls that were used. These include unstained, biological, isotype, fluorescence minus-one or negative controls present within the cell suspension. |
· We have explained about how the gating was performed in the methods section. · Changes are shown in red font. |
|
2) Results are often redundant and can be revised |
· We have reviewed the manuscript and revised as appropriate. · Changes are shown in red font. |
|
3) Flow cytometry data must be also displayed as histogram or dot plot. |
· Thank you for the comment. Figure 2 has been revised and is displayed as a histogram |
|
Results from qRT-PCR should be analyzed using the comparative DCt method |
· We have mentioned in the methods section that the data was normalized relative to the housekeeping gene. · Changes are shown in red font. |
|
Study limitation should be discussed |
· Study limitation has been mentioned in the conclusion · Changes are shown in red font. |
|
Figures are of poor quality |
· We have revised the figures |
Reviewer 2 Report
Sitti Rahma Abdul Hafid et al TRF found that TRF can be used as an adjuvant to induce activation and promote survival of T-cells, and thus contributed to inhibition of tumor growth in DC-vaccine immunotherapy tumor model. Moreover, the authors further identified SATB1 is a potential target. This is of great importance to the field. The experiments were well designed; however, I have the following concerns needed to be addressed:
Overall the manuscript is a lack of statistical analysis. Please add statistical analysis in Method section, as well as in the figure legend.
Figure 1, please run statistical analysis, and add in the figure legend the methods used.
Figure 3, 5 & 6, please add in the legend the statistical analysis used, and how many samples have been used to run the statistical analysis.
Figure 2 is confusing. The percentage should be “% out of total live cells” in certain group of animals. If that is the case, the total is not 100 for a certain marker (for example CD80:84%). Thus, pie chart is not appropriate, and column chart should be used instead. If percentage is defined differently, please make it clear.
For microarray study, how many replicates have been used?
Since the microarray analysis only generated 4 DEGs. If a looser condition, for example, p<0.05 & fold change >1.5fold, is used, more DEGs will be pop-up. It will be helpful to have that information.
in Figure 4c, the volcano plot for up- and down-regulated gene, the p value threshed was different. Why chose different threshed?
Author Response
Response to Comments by Reviewer 2
Sitti Rahma Abdul Hafid et al TRF found that TRF can be used as an adjuvant to induce activation and promote survival of T-cells, and thus contributed to inhibition of tumor growth in DC-vaccine immunotherapy tumor model. Moreover, the authors further identified SATB1 is a potential target. This is of great importance to the field. The experiments were well designed; however, I have the following concerns needed to be addressed:
|
Comments |
Response from authors |
|
Overall the manuscript is a lack of statistical analysis. Please add statistical analysis in Method section, as well as in the figure legend. |
· Thank you for the comment · We have revised the methods section to include statistical analysis · Information on statistical analysis have been included in the figure legend, where relevant · Changes are shown in red font |
|
Figure 1, please run statistical analysis, and add in the figure legend the methods used. |
· Information on statistical analysis used and the results have been included in the figure legend · Changes are shown in red font |
|
Figure 3, 5 & 6, please add in the legend the statistical analysis used, and how many samples have been used to run the statistical analysis. |
· Information on statistical analysis used and the results have been included in the figure legend · The number of samples used have been mentioned in the figure legend · Changes are shown in red font |
|
Figure 2 is confusing. The percentage should be “% out of total live cells” in certain group of animals. If that is the case, the total is not 100 for a certain marker (for example CD80:84%). Thus, pie chart is not appropriate, and column chart should be used instead. If percentage is defined differently, please make it clear. |
· Thank you for the comment. · We have revised Figure 2 and presented the data as histograms |
|
For microarray study, how many replicates have been used? |
· 3 replicates for each sample have been used for this study |
|
Since the microarray analysis only generated 4 DEGs. If a looser condition, for example, p<0.05 & fold change >1.5fold, is used, more DEGs will be pop-up. It will be helpful to have that information. |
· In the microarray analysis reported in this paper, we only looked at differentially expressed genes in tumors from two group i.e. DC+TL with DC+TL+TRF · We agree that more genes would have been identified if we used a less stringent conditions as you had suggested but we purposely used stringent conditions as we wanted to identify putative target molecules that we can do further work on. · Both the above information have been included in the revised manuscript · Changes are shown in red font |
|
in Figure 4c, the volcano plot for up- and down-regulated gene, the p value threshed was different. Why chose different threshed? |
· When confidence interval (CI) was set at 95% (P < 0.05), we identified 4 differentially expressed genes. However, when CI was increased to 99%, we managed to narrow down to one putative gene. This gene appears to be a good target for further work as both qPCR and Western blotting analysis support the microarray findings. · The above information have been explained in the manuscript. · Changes are shown in red font |
Round 2
Reviewer 1 Report
The manuscript has been improved, therefore it is now suitable for publication